# Signature of spin-triplet exciton condensations in LaCoO₃ at ultrahigh magnetic fields up to 600 T

**Akihiko Ikeda** [1,2] ✉, **Yasuhiro H. Matsuda** [1], **Keisuke Sato**[3], **Yuto Ishii**[1], **Hironobu Sawabe**[1], **Daisuke Nakamura** [1,6], **Shojiro Takeyama** [1] & **Joji Nasu** [4,5]

Bose-Einstein condensation of electron-hole pairs, exciton condensation, has been effortfully investigated since predicted 60 years ago. Irrefutable evidence has still been lacking due to experimental difficulties in verifying the condensation of the charge neutral and non-magnetic spin-singlet excitons. Whilst, condensation of spin-triplet excitons is a promising frontier because spin supercurrent and spin-Seebeck effects will be observable. A canonical cobaltite LaCoO₃ under very high magnetic fields is a propitious candidate, yet to be verified. Here, we unveil the exotic phase diagram of LaCoO₃ up to 600 T generated using the electromagnetic flux compression method and the state-of-the-art magnetostriction gauge. We found the continuous magnetostriction curves and a bending structure, which suggest the emergence of two distinct spin-triplet exciton condensates. By constructing a phenomenological model, we showed that quantum fluctuations of excitons are crucial for the field-induced successive transitions. The spin-triplet exciton condensation in a cobaltite, which is three-dimensional and thermally equilibrated, opens up a novel venue for spintronics technologies with spin-supercurrent such as a spin Josephson junction.

In condensed matters, novel phases emerge at boundaries of incompatible phases, where competing interactions and fluctuations trigger spontaneous symmetry breakings. Insulator state via the Bose-Einstein condensation of electron-hole pairs, excitonic insulator or exciton condensation, has been eagerly investigated over 50 years since its prediction[1], which emerges at the boundary region between semimetals and semi-conductors and their photo-excited states. However, compelling evidence has been lacking, due to the difficulty in verifying the condensation of charge neutral and spin-singlet excitons[2–6].

On the other hand, the condensation of the spin-triplet excitons is more beneficial to investigate thanks to observable magnetic phenomena such as spin-supercurrent[7,8] and spin Seebeck effect[9], which are attractive for potential applications in spintronics and quantum computing technologies. Condensation of spin-triplet bosons is of fundamental interest in analogy with the spin-triplet superconductivity[10,11] and superfluidity of ³He[12]. The condensation of spin-triplet excitons has been rarely considered in conventional studies of exciton condensations[2–6].

Recently, theories have predicted that spin-triplet exciton condensation is stable in perovskite cobaltites with strongly correlated electrons[13–15]. This is based on the fact that the spin-state degrees of freedom are inherent in the cobaltites [see Fig. 1a and b] due to the competing Hund's coupling and the crystal field splitting, which can also be viewed as the novel degrees of freedom of atomic size excitons [see Fig. 1c]. The spin-triplet exciton condensation is essentially the quantum hybridization of the distinct exciton states. The hybridization

[1]Institute for Solid State Physics, University of Tokyo, Kashiwa, Chiba 277-8581, Japan. [2]Department of Engineering Science, University of Electro-Communications, Chofu, Tokyo 182-8585, Japan. [3]National Institute of Technology, Ibaraki College, Hitachinaka, Ibaraki 312-0011, Japan. [4]Department of Physics, Tohoku University, Sendai, Miyagi 980-8578, Japan. [5]PRESTO, Japan Science and Technology Agency, Honcho Kawaguchi, Saitama 332-0012, Japan. [6]Present address: RIKEN Center for Emergent Matter Science (CEMS), Wako 351-0198, Japan. ✉e-mail: a-ikeda@uec.ac.jp

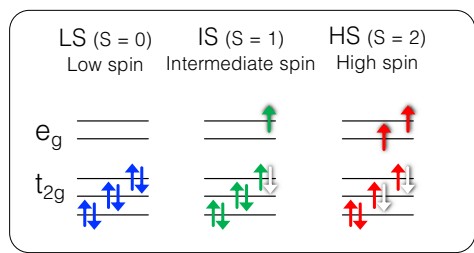

**b** Spin state degree of freedom in LaCoO$_3$

Correspondence

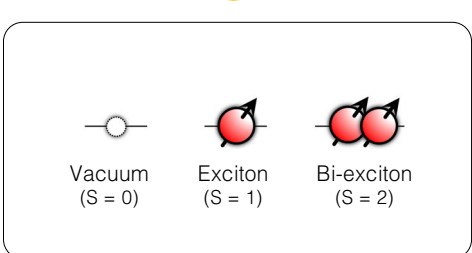

**c** Exciton degree of freedom in LaCoO$_3$

**Fig. 1 | Schematic illustrations describing the correspondence between spin-states and exciton states in LaCoO$_3$. a** Schematic crystal structure of LaCoO$_3$. **b** Schematic drawing of spin-state degrees of freedom of LaCoO$_3$. **c** Schematic drawing of excitonic degrees of freedom in LaCoO$_3$. LS and a pair of shaded electron and hole in (**b**) correspond to vacuum and an exciton in (**c**), respectively.

is due to the delocalization, interaction, and duality of excitons [see Fig. 2a] that are promoted when the energies of the vacuum and exciton states are in proximity by such external fields as magnetic fields [See Fig. 2b]. The LS and HS states correspond to the Mott insulator and the band insulator phases, respectively[13]. Thereby, the spin-triplet exciton condensation occurs at the boundary region between the incompatible phases [See Fig. 2b].

A doped perovskite cobaltite Pr$_{0.5}$Ca$_{0.5}$CoO$_3$ and its family compounds are the first candidate for the spin-triplet exciton condensation, exhibiting an insulating and paramagnetic ground state at low temperatures[16,17]. The origin of the insulator phase is, however, not completely uncovered because the significant valence transition of Pr complicates the phase transition. Next, the most well-known cobaltite LaCoO$_3$ has been claimed to be a promising candidate when placed under very high magnetic fields exceeding 100 T[18,19]. LaCoO$_3$ is an archetypal compound having a variety of spin-states, such as low-spin (LS: $S = 0$, $t_{2g}^6 e_g^0$), intermediate spin (IS: $S = 1$, $t_{2g}^5 e_g^1$) and high spin (HS: $S = 2$, $t_{2g}^4 e_g^2$) states [See Fig. 1b], which are viewed as vacuum, an exciton, and bi-exciton states, respectively [See Fig. 1c]. The spin-triplet exciton condensation will emerge when the magnetic field eliminates the spin-gap and changes the vacuum state (LS) to a magnetic exciton state (IS) or bi-exciton state (HS) [See Fig. 2b].

However, the experimental investigation has been challenging due to the need for very high magnetic fields beyond 100 T which is necessitated by the large spin-gap of ~100 K. Besides, one needs to probe the exciton states at such high magnetic fields. As solutions to these difficulties, recently, we reported the generation of 1200 T with electromagnetic flux compression (EMFC) method using a newly installed capacitor banks system in ISSP[20]. Furthermore, we succeeded in implementing a state-of-the-art high-speed 100 MHz strain gauge using fiber Bragg grating (FBG) and optical filter method[21], which have enabled us to measure magnetostriction in the $\mu$s-pulsed 1000 T environment. Magnetostriction is a crucial probe of the exciton states because the density of excitons is coupled to the lattice volume of

LaCoO$_3$, where exciton and bi-exciton states have larger ionic volumes with the occupation of $e_g$ orbitals that is spatially more extended than $t_{2g}$ orbitals as can be seen in the correspondence of spin-states and exciton states [See Fig. 1b, c.] Solidifications and Bose-Einstein condensations of excitons result in plateaus and slope of exciton density[18,19,22]. Thus, we expect that they also result in plateaus and slope in magnetostriction curves at very low temperatures, which is schematically shown in Fig. 2c. Note that the correspondences are analogs to the magnetization plateaus and slopes in magnon solids and superfluids, respectively[23,24].

Previously in LaCoO$_3$, the field-induced phase transitions are uncovered below 30 K ($\alpha$ phase)[25-27] and above 30 K ($\beta$ phase), which extend beyond 100 T, using magnetization and magnetostriction measurements up to 200 T generated by single turn coil (STC) method[28,29]. The two phases are considered superlattice formation (i.e., solidification) of the excitons, bi-excitons, and vacuum based on the observation of plateaux of magnetostrictions. Considering that the magnetization below 200 T is far from the full polarization of the spin-states, further orderings of excitons are expected to appear above 200 T. So far, only one magnetization study up to 500 T[30] has been reported. Now, we are fully equipped for the investigation of the evolution of the excitonic states in LaCoO$_3$ beyond 200 T.

## Results

Figure 3a–h show the results of magnetostriction measurements of polycrystalline LaCoO$_3$ up to 600 T with the initial sample temperatures $T_{ini}$ = 5, 78, 108, and 185 K. Figure 3a–d show the magnetostriction data and magnetic fields as a function of time. Magnetostriction data as a function of magnetic fields are shown in Fig. 3e–h. The magnetostriction data up to 150 T in ref. [29] are shown for comparison using green curves in Fig. 3f–h. The transition points are indicated in Fig. 3e–h and are summarized using star-shaped symbols in Fig. 3i along with the previous results up to 150 T[28,29] shown using filled and open triangles and circles. The transitions and features below 200 T well reproduce the previous observations up to 150 T [See Fig. 3e–h]

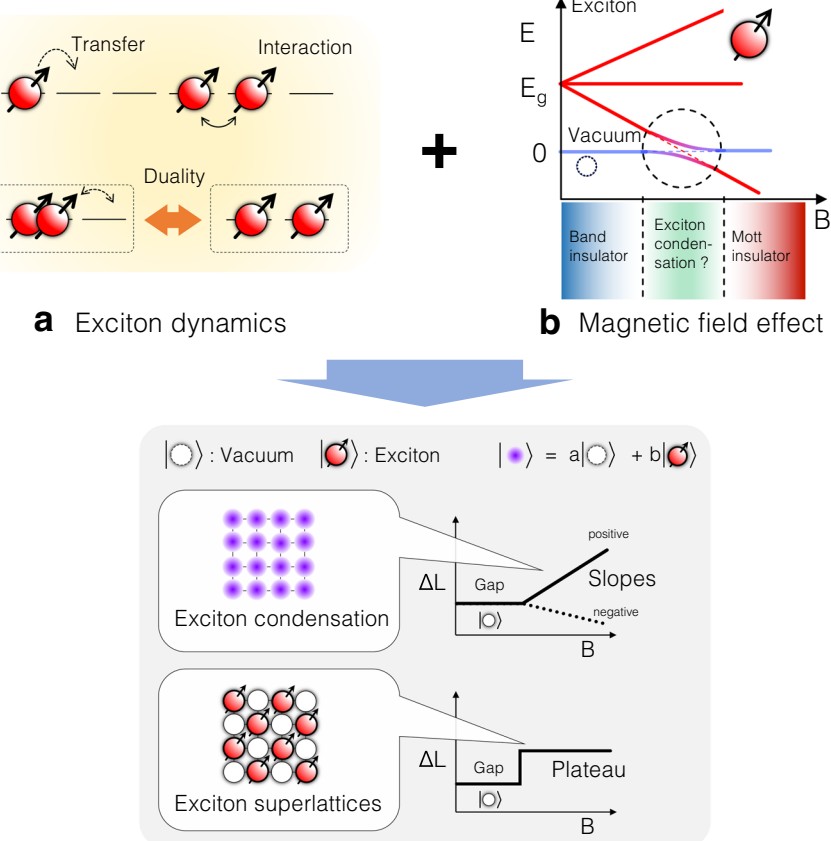

**a** Exciton dynamics

**b** Magnetic field effect

**c** Emergent phases and strain effects under magnetic fields

**Fig. 2 | Schematic illustrations describing the possible occurrence of exotic orderings of excitons in LaCoO₃ under high magnetic fields. a** Schematic drawing of the interactions and dynamics of the excitons. **b** Schematic drawing of the magnetic field effect on the exciton states in LaCoO₃. **c** Expected emergence of the magnetic field induced phases in LaCoO₃, and the resultant magnetostriction curves.

where the transition fields are denoted as $B_{C1}$ and $B_{C2}$. The propagation of a shock wave inside the sample initiated by the strong transitions of $B_{C1}$ and $B_{C2}$ are present as indicated in Fig. 3a–d, which is also reported in refs. [29,31]. We distinguish the intrinsic features and the shock propagation by carefully observing the sound speed and temperature dependence. See Supplementary Note 1 for the details. The temperature changes of the sample during the application of the $\mu$-second pulsed magnetic field are considered significant at $T_{ini} = 5$ K, while it is negligible at $T_{ini} > 30$ K based on the sufficiently large heat capacity above 30 K[32].

The magnetostriction plateau of the $\alpha$ phase at $T_{ini} = 5$ K that appears above 70 T is found to persist up to 600 T, revealing its significant stability under magnetic fields. On the other hand, the $\beta$ phase is found to show a negative slope of magnetostriction [See slope 1 in Fig. 3g] between 170 T and 380 T. Furthermore, we find a bending structure indicating a new phase transition at 380 T denoted as $B_{C3}$ from the $\beta$ phase to a novel state in the data at $T_{ini} = 78$ K. The new state is termed the $\gamma$ phase, which is characterized by the positive slope beyond 380 T [See slope 2 in Fig. 3g]. The sharp slopes of slope 1 and slope 2 at $T_{ini} = 78$ K become smeared in the data obtained at $T_{ini} = 108$ K as shown in Fig. 3f. Thus, the sharp slopes of slope 1 and slope 2 obtained at $T_{ini} = 78$ K are not thermal origins. But rather, we argue that they originate in quantum fluctuations of excitons such as exciton condensations as schematically shown in Fig. 2c. Previously in ref. [29], the $\beta$ phase was falsely considered a plateau due to the measurement regions limited below 200 T. Furthermore, the negative slope itself disappears at the $T_{ini} = 185$ K data as shown in Fig. 3e, while the feature of the transition of $B_{C2}$ remains. These features indicate that both $\beta$ and

$\gamma$ phases become unstable at ~100 K, that the $\gamma$ phase is completely absent at ~185 K, and also that the feature of the $\beta$ phase does not completely disappear at ~185 K.

## Discussion
Here, we discuss the origins of the $\alpha$, the $\beta$, and the $\gamma$ phases in LaCoO₃ induced at ultrahigh magnetic fields. We claim that the $\alpha$ phase is a superlattice of spin-triplet excitons and vacuum, based on the present observation of the magnetostriction plateau at $T_{ini} = 5$ K shown in Fig. 3h and that the magnetization of $\alpha$ phase is only 1/8 or 1/4 of the saturation of the exciton states[25,26]. In the exciton superlattice, the number of the exciton is discrete[15], leading to the magnetostriction plateau as is shown in the data at 5 K in Fig. 3h. This is in good agreement with the previous observations and considerations on the $\alpha$ phase[25–27,29]. The remarkable stability of the $\alpha$ phase that sustains up to 600 T suggests the localized nature of the spin-triplet excitons in the $\alpha$ phase of LaCoO₃.

On the other hand, the $\beta$ and the $\gamma$ phases show magnetostriction slopes denoted as slope 1 and slope 2 in Fig. 3f, g. Based on the observation that they are nonthermal in origin, they indicate that exciton density varies continuously as a function of magnetic fields. Considering the firm coupling of lattice volume and exciton density in LaCoO₃, these behaviors imply the continuous change of lattice volume as a function of magnetic fields, which is a characteristic of Bose-Einstein condensation of excitons[18,19,22]. In a condensate of excitons, the single site wave function of Co is described by the linear combination of $|\phi_{EC}\rangle = a|0\rangle + b|1\rangle + c|2\rangle$, where $|0\rangle$, $|1\rangle$, $|2\rangle$ represents the vacuum, an exciton, and a bi-exciton state, respectively. A global

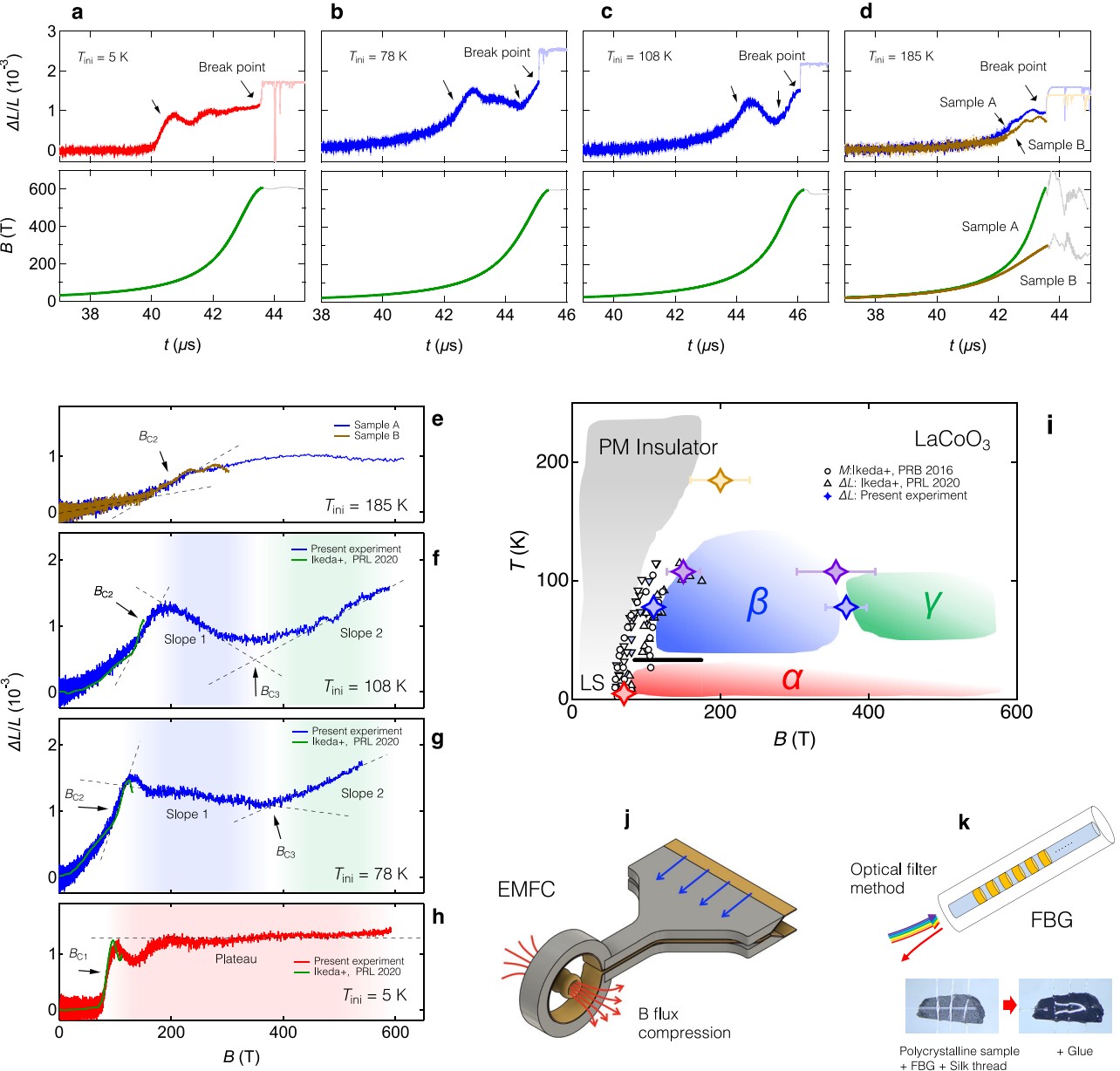

**Fig. 3 | Magnetostriction data up to 600 T for LaCoO$_3$ at various temperatures and the summarizing phase diagram. a-d** Result of magnetostriction measurement of LaCoO$_3$ shown as a function of time, along with the magnetic field profiles generated using the EMFC method. The initial sample temperatures $T_{ini}$ are varied from 5 K to 185 K. **e-h** Magnetostriction data of LaCoO$_3$ shown as a function of magnetic field at various $T_{ini}$, along with the reference data (Green curves) previously obtained using STC[29]. **i** Summary of the magnetostriction measurement of LaCoO$_3$ on a magnetic field-temperature plane up to 600 T along with the previously obtained data using magnetization[28] and magnetostriction measurements[29]. Sample A and Sample B in (**d**) and (**e**) represent the data of LaCoO$_3$ samples positioned at the center (sample A) and 7 mm off the center (sample B) of the coil in the axial direction. **j** An illustration of the main coil and liner during the EMFC. **k** Illustrations of the FBG fiber and sample glued to the FBG fiber used in the experiment.

wave-function can be expressed as $|\psi_{EC}\rangle = \prod_i |\phi_{EC}\rangle_i$. The continuous change of magnetostriction and magnetization is realized because the coefficients $a, b, c$ can change continuously in exciton condensates as a function of magnetic field[18,19]. Another possible origin of continuous magnetostriction is thermal activation. We rule out this possibility based on the negative magnetostriction observed in slope 1 of Fig. 3g. No negative thermal expansion is reported in LaCoO$_3$[33]. The possibilities of orbital order and domain re-orientations are raised based on the facts that LaCoO$_3$ has an orbital degree of freedom in IS and HS states and that it is a slightly anisotropic pseudo-cubic system[34]. We rule out these possibilities because we used polycrystalline samples.

It is well known that the exciton condensations are difficult to be evidenced by a single experiment. One needs various macroscopic and

microscopic experiments[14]. Here, we tentatively construct a phenomenological model Hamiltonian just for a qualitative discussion of the possible appearance of multiple exciton condensations in LaCoO$_3$ at ultrahigh magnetic fields. We start from the five-orbital Hubbard model with six electrons per site for the electronic states of Co ions. In the strong correlation limit, there are contributions from the exchange processes of $e_g$ and $t_{2g}$ electrons between neighboring Co ions. This leads to the itinerancy of excitons ($\mathcal{H}_{trans}$) and the fusion of two excitons to a bi-exciton ($\mathcal{H}_{dual}$). The model introduced here consists of these two quantum processes in addition to the local exciton energy with the Zeeman field ($\mathcal{H}_{loc}$) and classical interaction between neighboring (bi-)excitons ($\mathcal{H}_{int}$). Note that the present model includes the bi-exciton state, which is in contrast to the previous literature on exciton

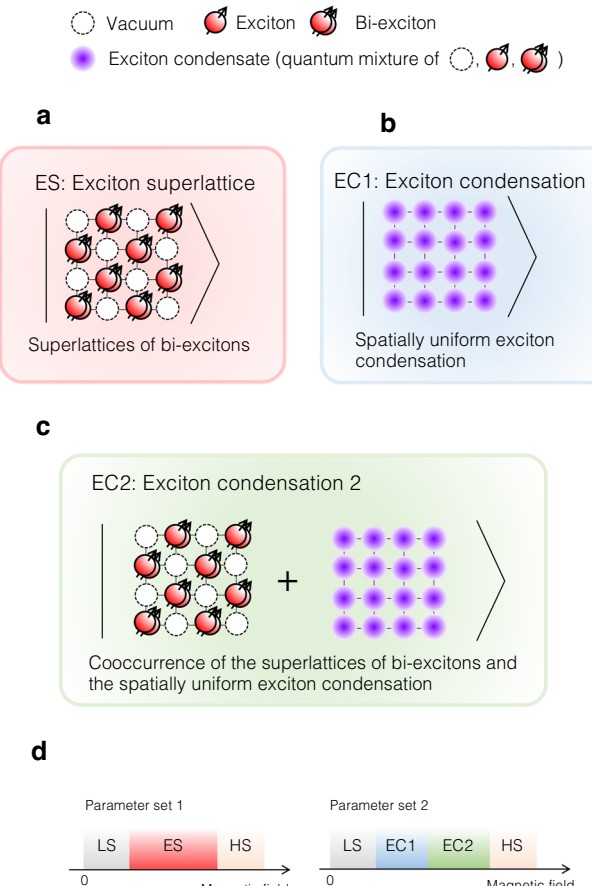

**Fig. 4 | Schematic illustrations of calculated excitonic phases. a** Exciton superlattice (ES), where exciton and vacuum form a superlattice. **b** Exciton condensation 1 (EC1), where Bose-Einstein condensation of excitons occur spatially uniformly. **c** Exciton condensation 2 (EC2), where Bose-Einstein condensation of excitons occur with a translational symmetry breaking. **d** Magnetic field dependence of the mean-field phases with the assumption of distinct parameter sets. See Supplementary Note 2 for the details of the parameters.

condensation, where only the exciton state is taken into account in refs. [18,19]. Note also that the model is a quantum upgrade of the classical model introduced in ref. [26]. The details are given in Supplementary Note 2.

Here, we just show the essence of the calculated results in Fig. 4a–d, for simplicity. When the contributions from the quantum processes are weak, the phase transition occurs from the vacuum ground state to an exciton superlattice (ES) as depicted in Fig. 4a with a plateau of exciton density. In the ES phase, the exciton state does not change with increasing magnetic fields, which would result in a plateau of magnetostriction. ES is a possible candidate for the $\alpha$ phase. Further, with sufficient contribution from the duality term, the calculated results show that the phase transitions occur from the vacuum ground state to two kinds of exciton condensations with increasing magnetic fields. As schematically shown in Fig. 4b, the exciton condensation 1 (EC1) is a spatially uniform exciton condensation made of a mixture of all the exciton states as $|\phi_{EC1}\rangle = a|0\rangle + b|1\rangle + c|2\rangle$. EC1 is followed by the appearance of exciton condensation 2 (EC2) with increasing magnetic fields. In EC2, both order parameters of exciton superlattice and exciton condensation become finite simultaneously as schematically shown in Fig. 4c. In EC1 and EC2, the exciton states change continuously with increasing magnetic fields, which would result in slopes of magnetostriction. EC1 and EC2 are possible candidates for the $\beta$ and $\gamma$ phases. As shown in Fig. 4d, EC1 and EC2 appear with a different parameter set from that showed ES. The large lattice change between

the $\alpha$ and $\beta$ phase[29] can be an origin of such variations of parameter sets in LaCoO$_3$ at above 100 T.

Lastly, we discuss the result in light of the spin-crossover physics in LaCoO$_3$. We argue that the HS-IS duality and the competing parameters are of crucial importance when considering the controversial properties of LaCoO$_3$. It has been a long-standing controversy for over half a century how to understand the peculiar temperature evolution of magnetic and electric properties of LaCoO$_3$ in terms of the spin-states. The involvement of various spin-states LS, IS, and HS, and also their interactions have been proposed[35]. The controversy has been long-lasting largely due to the fact that no long-range order has been found in the temperature evolution of LaCoO$_3$[35]. The present study and previous papers[26,29] have reported that various ordered phases emerge under ultrahigh magnetic fields in LaCoO$_3$. However, previously reported models can not account for the presently observed complex phase diagram (Fig. 3i) and especially the continuous change of the magnetostriction in the $\beta$ and the $\gamma$ phases. This indicates that the HS-IS duality, which is included in the present model, is an important factor in LaCoO$_3$. Further microscopic experiments such as X-ray diffraction[36] and thorough theoretical investigations are mandatory to clarify this point.

## Methods
The magnetic fields up to 600 T are generated using the electromagnetic flux compression method[20] in the Institute for Solid State Physics, the University of Tokyo, Japan. The magnetostriction measurement is performed using our state-of-the-art high-speed strain gauge utilizing fiber Bragg grating and optical filter method[21]. Polycrystalline samples of LaCoO$_3$ were prepared by the solid-state-reaction of predried La$_2$O$_3$ and CoO at 1300 C˚ in air. The sample is cut out in $1 \times 3 \times 0.5$ mm. FBG fiber is glued tightly onto a specimen of the sample using Stycast 1266.

## Data availability
The data that support the findings of this study are available in Zenodo with the identifier [https://doi.org/10.5281/zenodo.7627257][37].

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

## Acknowledgements

AI is financially supported by MEXT Leading Initiative for Excellent Young Researchers Grant No. JPMXS0320210021, JSPS KAKENHI Grant-in-Aid for Early-Career Scientists Grant No. 18K13493, and Basic Science Program No. 18-001 of Tokyo Electric Power Company (TEPCO) memorial foundation.

## Author contributions

A.I. designed the research. K.S prepared the sample. A.I., Y.H.M., H.S., Y. I., D.N., S.T. performed the experiments. A.I. analyzed the data. J.N. and A.I. developed the model. J.N. performed the calculations. A.I, Y.H.M., J.N. discussed the results. A.I. wrote the manuscript with input from all co-authors.

## Competing interests
The authors declare no competing interests.
