## [Peer Review File · Nature Communications]

REVIEWER COMMENTS

Reviewer #1 (Remarks to the Author):

The present article is devoted to a long-standing problem of LaCoO₃ magnetic phase diagram. The authors have determined the magnetic phase diagram up to 600 T and revealed three high-field phases, two of which contained Bose-Einstein condensate.

In this work, for the first time, magnetostrictive measurements were carried out in an electromagnetic flux compression (EMF) facility. It should be mentioned that the EMF method provides the highest laboratory magnetic fields. On the other hand, a choice of measurement techniques in these devices is very poor because of hard experimental conditions (short duration of the magnetic field, huge time derivative of the magnetic flux density, intensive electromagnetic interferences). Therefore, an introduction of a new experimental measurement technique is of great importance.

I find this work quite exciting. To my opinion, the paper is suitable for publication in Nat. Comm. Nevertheless, I have got few questions on the experimental technique:

1. There are shifts in time diagram in Fig.3 (I mean the B(t) dependences at different temperatures). Are they occasional or they reflect different operational modes of the facility?
2. One can see damped oscillations at the plateau in Fig.3a (T_{ini}=5 K). Are they related to natural mechanical vibrations of the sample? If so, the plot in Fig.3a (T_{ini}=108K) could be also interpreted as natural vibrations. Can the authors exclude this possibility? This is important to prove robustness of the conclusions.
3. Can an axial magnetic field gradient in the experimental facility produce a sizable axial force on the sample (or strain in the sample)?

There is also a question on the theoretical section of the work:

4. Is any uniform state of itinerant excitons the Bose-Einstein condensate? If not, how can one distinguish between the Bose-Einstein condensate and non-condensed itinerant bosons? Maybe this question looks naïve, however an elucidation of this point would improve accessibility of the paper to a broad audience.

There are misprints in the titles of the left and right panels in Fig.4 (“Exciton condensation 1 (2)”).

Reviewer #3 (Remarks to the Author):

The manuscript experimentally investigated the magnetostriction of LaCoO_3 under magnetic fields as high as 600 T, with the aim of finding evidence for spin-triplet exciton condensation. The authors also proposed a theoretical model to further support their findings. However, the data presented do not convince me that they have achieved this goal. Therefore, I cannot recommend its publication in Nature Communications, at least not in its current version. My concerns are as follows:

1) Where is Fig. 3d? Without it, I cannot understand the results shown in the paragraph spanning pages 3 to 4 at all. In particular, what does "*gapless magnetostriction slopes*" mean? How did the authors arrive at "*These gapless behaviors evoke the Bose-Einstein condensation of spin-triplet excitons*"?

2) Figure 2c shows the theoretical correspondence of the experimental work. The authors expected to verify the exciton condensate from observing such "Slope" and "Plateau" behaviors. Given the importance of this theoretical basis and the distinctly different backgrounds of the readers of Nature Communications, it would be better for the authors to elaborate on its relationship with the spin-triplet exciton condensation in more detail, rather than simply provide three references [14, 15, 26]. In my opinion, this hinders the readability and continuity of the manuscript. What makes me more confused is that there seems to be no work on this aspect in the references [14, 15, 26]. The word magnetostriction did not appear in any of the three publications. I wonder if the authors submitted the wrong version of the manuscript?

3) In the last part, the authors wrote down a complex Hamiltonian with eight parameters. Since the model does not include temperature explicitly, it is natural that it should be more suitable for the low temperature case like the α phase. But, as the authors pointed out, it fails to reproduce the α phase. So how do we understand the physics that this model gives us? With eight parameters, how to tell whether it is the result of tunable parameters or it really captures the essence of physics?

4) There are several typos.

Referee report to

Authors:

A. Ikeda, Y.H. Matsuda, K. Sato, Y. Ishii, H. Sawabe, D. Nakamura, S. Takeyama, and J. Nasu

Title:

“Spin triplet condensations in LaCoO₃ at ultrahigh magnetic fields up to 600 T”

Spin state crossover processes induced by changes in the electronic configuration of transition metal ions dramatically change physical properties of solids.

Thus, these processes are also considered as new spin-state degree of freedom and have been studied in detail for more than 50 years. Among other compounds, LaCoO₃ in particular is the most analysed substance in which spin-crossover occurs. In addition to resulting effects such as conductivity changes, magnetization jumps, etc., magnetostriction measurements have been carried out because the electronic states strongly depend on and are influenced by the Co-O and Co-Co bond lengths. More than 100 publications prove the great interest in the spin-crossover problem, which Ikeda et al. also address in the present draft.

In the meantime, there is a consensus that the pure existence of LS-IS-HS states is not sufficient for the description and that a more complex scenario exists. From a large number of experiments and theoretical simulations, LS-HS coexistence with temperature- and magnetic field-dependent distribution is derived in contrast to a pure IS story. The present work, like some modern papers, links the problem to exciton condensation or the formation of exciton superlattices. In any case, the performance of magnetostriction measurements up to 600 T using the FBG method and electromagnetic flux compression is new and shows a great experimental experience. These measurements are currently only possible worldwide at the ISSP. But, it should be noted here that measurements (magnetization) at LaCoO₃ up to 500 T have already been performed in 2012 (Platonov et al. Phys. of the Solid State 54 (2012) 279). The excellent measurements of the ISSP team are complemented in the second part of the manuscript by a mean-field calculation, which obviously originates from another group and attempts to reproduce the experimental results and describe the possible processes in LaCoO₃ by extending the approaches used in other models (e.g. Altarawneh PRL 109 (2012) 037201) with exciton delocalisation and hybridization terms.

Specifically, the following question arises:

- 1) Ikeda et al. show in their results a change in the slope of the magnetostriction curves at about 400 T and deduce from this the existence of a new gapless phase (γ) in the phase diagram between 50 K and about 100 K. According to general statements (compare also Fig. 1), however, the gapless phase used for the description is characterised by a positive slope of the expansion curve. Furthermore: The upper limit of the γ phase at approx. 100 K is not proven, since measurements at higher temperatures are not available. Measurements at higher temperatures would be helpful to confirm the B_{c3} transition. And: The reduction of the sharpness of B_{c3} at 108 K is rather due to the higher temperature
- 2) The measurement results are combined with a mean-field calculation (in comparison to previous model calculations, e.g. Sotnikov et al. Sci. Rep. 6 (2016) 30510 extended). In addition to the well-known phases SSO and EC (spin-state order or exciton condensation), new phases are postulated (ES becomes ES1 and ES2, possibly connected to β and γ) and shown by spin numbers in Fig. 4. The energy curves in Fig. 4, however, show only the states $|0.0\rangle$, $|0.2\rangle$ and $|2.2\rangle$ as states of the lowest energy. But, this field dependent scenario would be already known.
- 3) Why the transition from EC1 to EC2 should be of second order, as later discussed in the text, is also not clearly justified.
- 4) The assignment of the calculated phases LS, EC1 and EC2 to α , β and γ and the real magnetic field to the "normalized field" seems arbitrary ("may correspond", "we speculate"). Some more concrete (quantitative) results of the simulation would make the connection between Figs. 3 and 4 more clear.
- 5) Some formal errors are: Fig. 3(d) does not exist. "Isotropic" on page 4.

Overall, I have to come to the conclusion that the results presented are very valuable in the context of the long-lasting discussion due to the new suggestions and ideas and, especially the experimental results, should definitely be published. However, it must also be stated that the proposed conclusions bring only little clarification of the basic problem and do not resolve the existing controversies (even the possible LS-IS-HS hybridizations already discussed are not clarified any more than elsewhere).

Reviewer #2 (Remarks to the Author):

The manuscript is clearly written and linguistically correct, the illustrations instructive and of very good quality. However, the connection between experiment and theory does not yet seem to be fully harmonized. Because of this fact and because I cannot see enough significant new aspects for clarification on the basic problem of the spin-crossover in LaCoO_3 , I do **not want to recommend the publication in nat.comm** at the present time.

Comments from reviewer #1 - 101

The present article is devoted to a long-standing problem of LaCoO₃ magnetic phase diagram. The authors have determined the magnetic phase diagram up to 600 T and revealed three high-field phases, two of which contained Bose-Einstein condensate. In this work, for the first time, magnetostrictive measurements were carried out in an electromagnetic flux compression (EMF) facility. It should be mentioned that the EMF method provides the highest laboratory magnetic fields. On the other hand, a choice of measurement techniques in these devices is very poor because of hard experimental conditions (short duration of the magnetic field, huge time derivative of the magnetic flux density, intensive electromagnetic interferences). Therefore, an introduction of a new experimental measurement technique is of great importance.

I find this work quite exciting. To my opinion, the paper is suitable for publication in Nat. Comm. Nevertheless, I have got few questions on the experimental technique:

Our reply

We appreciate the positive comment.

Comments from reviewer #1 - 102

1. There are shifts in time diagram in Fig.3 (I mean the $B(t)$ dependences at different temperatures). Are they occasional or they reflect different operational modes of the facility?

Our Reply

Fig. 3 a-d (main text) Bottom panels show magnetic field curves.

Yes, they reflect the operational conditions of the experimental setup.

Please see Fig. 3 a-d (partly) shown above. As reviewer #1 pointed out, there are variations of the B field curves as shown in Fig 3a-d. especially the time when the B_{max} (t_{bmax}) is reached. The results and the discharge conditions are summarized in the Table below.

	Fig 3a	Fig 3b	Fig 3c	Fig 3d
t_{bmax}	43.5 μ s	45.5 μ s	46.2 μ s	43.5 μ s
$E_{discharged}$	2.56 MJ	2.24 MJ	1.92 MJ	2.24 MJ
$V_{charged}$	40 kV	40 kV	40 kV	40 kV
$C_{discharged}$	3.2 mF (8/8 modules)	2.8 mF (7/8 modules)	2.4 mF (6/8 modules)	2.8 mF (7/8 modules)

Table. discharge parameters of electromagnetic flux compression system

The variation is largely due to the variation in the number of capacitor banks used. As you see in the bottom row in the Table, the larger the total energy, the earlier the B_{max} is reached.

We note that the variation of the total energy arises from the accidental failure of the synchronization of the energy discharge of the capacitor banks. In addition, we note that B_{max} is practically determined by when the B dot sensor is broken, which is not always the same as the real B_{max} .

Comments from reviewer #1 - 103

2. One can see damped oscillations at the plateau in Fig.3a ($T_{ini}=5$ K). Are they related to natural mechanical vibrations of the sample? If so, the plot in Fig.3a ($T_{ini}=108$ K) could be also interpreted as natural vibrations. Can the authors exclude this possibility? This is important to prove robustness of the conclusions.

Our Reply

Fig. 3 a-d (main text): Top panels show magnetostriction curves up to 600 T

We exclude the possibility of mechanical vibration as a source of the oscillating feature in 108 K data (Fig. 3c) on the following basis.

The key idea is temperature dependence. If the oscillating feature in 108 K data is actually a vibration, it should have appeared in the 78 K data with similar amplitude and frequency. We see only a small oscillating feature in the 78 K data above B_{c2} . The sharp slope features in 78 K are more apparent. 108 K is only 30 K above 78 K. The mechanical property is similar in this temperature change as reported in an ultrasound study [T. Sin Naing, et al, J. Phys. Soc. Jpn. **75**, 084601 (2006)]. Similarly, at the data 185 K, no negative slope is observed. This also indicates that the negative slopes of the data at 78 K and 108 K are intrinsic features.

Quantitatively, If the feature in 108 K is vibration, the shock propagation speed corresponds to 1.5 km/s with 250 kHz vibration and a sample size of 3 mm. In the same way, we obtain 3 km/s with 600 kHz vibration and a sample size of 3 mm. According to [T. Sin Naing, et al, J. Phys. Soc. Jpn. **75**, 084601 (2006)], the sound velocity change from 5 K to 108 K is 15%. Thus the oscillating feature in 108 K data is too slow for a shock propagation vibration.

We described the above argument in Section I of the supplemental material and put the reference in the main manuscript as follows.

is also reported in Refs. [16, 31]. We distinguish the intrinsic features and the shock propagation by carefully observing the sound speed and temperature dependence. See Supplemental Material for the details [32]. The temperature changes

The revised part of the main text which appears in page 2

Comments from reviewer #1 - 104

3. Can an axial magnetic field gradient in the experimental facility produce a sizable axial force on the sample (or strain in the sample)?

Our Reply

As the reviewer indicated the spatial variation of the B field becomes severe especially when the B_{max} is obtained. We used samples of 3 mm in axial length. It is ±1.5 mm from the B field center. The axial variation of B in EMFC was reported in [D. Nakamura et al., Rev. Sci. Instrum. 85, 036102 (2014).] as shown in the figure below. The axial field variation is small (<20 T) up to 500 T. It gets significant (>50 T) above 600 T. The study was performed on our previous EMFC system, whose record B_{max} is about 700 T [S. Takeyama et al., J. Phys. D 44, 425003 (2010)].

In the present study, we used a new EMFC system whose record B_{max} is 1200 T [D. Nakamura (2018)] at the compression energy of 3.2 MJ. In the present measurements, we used smaller compression energy of 1.9 - 2.5 MJ to reduce the B_{max} down to 600 T. Using the smaller compression energy, one obtains a lower B_{max} with better B field variation. Though we have not directly measured the axial distribution of B-field in the current EMFC system, we estimate that the B-field variation is similar to the reported one at $t = 39.7 \mu\text{s}$ in the bottom figure below because we have sufficiently reduced the B_{max}, where the 20 mm plateau of the B-field should be realized due to the relaxed B_{max}. Thus, in the present study, the sample size is negligibly small compared to the field variation in the axial direction. We note that we plan to measure the B-field distribution in the future for the machine study.

D. Nakamura et al., Rev. Sci. Instrum. 85, 036102 (2014)

Figs: Measurement of the axial distribution of the magnetic field in EMFC

Comments from reviewer #1 - 105

There is also a question on the theoretical section of the work:

4. Is any uniform state of itinerant excitons the Bose-Einstein condensate?

Our Reply

It is true at zero temperature within the mean-field theory. It may not hold, however, at finite temperatures and when strong quantum fluctuation exists. BEC-type excitonic insulators are known to form preformed excitons at high temperatures, which then condensate at low temperatures where a complete coherence is obtained [Phys. Rev. B 90, 155116 (2014)]. On the other hand, at high temperatures the coherence of the excitonic condensate is lost, where excitons are itinerant without BEC, which is a preformed excitonic state. Recently, theoretical researchers argue preformed excitonic liquid, that is something like imperfect excitonic condensate [Phys. Rev. B 90, 115146 (2014)].

If not, how can one distinguish between the Bose-Einstein condensate and non-condensed itinerant bosons? Maybe this question looks naive, however an elucidation of this point would improve accessibility of the paper to a broad audience.

Our Reply

Theoretically, BEC can be detected by observing that the order parameter such as $\langle b \rangle$ is nonzero. On the other hand, experimentally, the order parameter is not considered to be directly observable [J. Kunes, J. Phys.: Cond. Mat. 27, 333201(2015)]. Thus, experimental verification is achieved only by collecting collateral evidence. Observation of microscopic and macroscopic properties like magnetic excitations, lattice changes, thermal properties, and magnetic supercurrent will be a great help. The present study focuses on the condensation of spin-triplet excitons. Thus, magnetic properties can also be a great help. The present magnetostriction study is the first attempt at such a study.

Comments from reviewer #1 - 106

There are misprints in the titles of the left and right panels in Fig.4 (“Exciton condensation 1 (2)”).

Our Reply

They are now corrected. Thank you.

Corrected Fig 4d-g is shown above

Comments from reviewer #2 - 201

Spin state crossover processes induced by changes in the electronic configuration of transition metal ions dramatically change physical properties of solids.

Thus, these processes are also considered as a new spin-state degree of freedom and have been studied in detail for more than 50 years. Among other compounds, LaCoO₃ in particular is the most analyzed substance in which spin-crossover occurs. In addition to resulting effects such as conductivity changes, magnetization jumps, etc., magnetostriction measurements have been carried out because the electronic states strongly depend on and are influenced by the Co-O and Co-Co bond lengths. More than 100 publications prove the great interest in the spin-crossover problem, which Ikeda et al. also address in the present draft.

In the meantime, there is a consensus that the pure existence of LS-IS-HS states is not sufficient for the description and that a more complex scenario exists. From a large number of experiments and theoretical simulations, LS-HS coexistence with temperature- and magnetic field-dependent distribution is derived in contrast to a pure IS story. The present work, like some modern papers, links the problem to exciton condensation or the formation of exciton superlattices. In any case, the performance of magnetostriction measurements up to 600 T using the FBG method and electromagnetic flux compression is new and shows a great experimental experience. These measurements are currently only possible worldwide at the ISSP. But, it should be noted here that measurements (magnetization) at LaCoO₃ up to 500 T have already been performed in 2012 (Platonov et al. Phys. of the Solid State 54 (2012) 279).

Our Reply

We appreciate the positive comment.

The paper 2012 (Platonov et al. Phys. of the Solid State 54 (2012) 279) is now cited.

orderings of excitons are expected to appear above 200 T. So far, only one magnetization study up to 500 T [30] has been reported. Now, we are fully equipped for the investigation of

The excellent measurements of the ISSP team are complemented in the second part of the manuscript by a mean-field calculation, which obviously originates from another group and attempts to reproduce the experimental results and describe the possible processes in LaCoO₃ by extending the approaches used in other models (e.g. Altarawneh PRL 109 (2012) 037201) with exciton delocalization and hybridization terms.

Specifically, the following question arises:

Our Reply

We appreciate the positive comment.

The paper [Altarawneh PRL 109 (2012) 037201] is now cited, also in the theoretical section. See the figure below.

The Hamiltonian can be considered as a quantum upgrade of the classical exciton model introduced in Ref. [28], where the successive formation of exciton superlattices is discussed. The

Comments from reviewer #2 - 202

1) Ikeda et al. show in their results a change in the slope of the magnetostriction curves at about 400 T and deduce from this the existence of a new gapless phase (γ) in the phase diagram between 50 K and about 100 K. According to general statements (compare also Fig. 1), however, the gapless phase used for the description is characterized by a positive slope of the expansion curve.

Our Reply

The part of revised Fig. 2c is shown above

As the reviewer stated, we have simply indicated in Fig. 2c that a positive slope infers the excitonic condensations. Now, we have revised Fig. 2c so that a negative slope can also infer excitonic condensation. We argue that a “Slope” is fundamentally important because of its continuous change of lattice, regardless of its sign. The slope indicates the finite compressibility, which is one characteristic of exciton condensation.

Furthermore: The upper limit of the γ phase at approx. 100 K is not proven, since measurements at higher temperatures are not available. Measurements at higher temperatures would be helpful to confirm the B_{C3} transition. And: The reduction of the sharpness of B_{C3} at 108 K is rather due to the higher temperature

Our Reply

We have performed another EMFC experiment at a sample temperature of 185 K. The results are shown in Figs. 4d, 4e, and 4i. The EMFC experiment has been carried out with the simultaneous ΔL measurement of two samples that are located at the field center and 7 mm away from the center. As you can see in Fig. 4e the results from the 2 samples are identical up to 300 T, where the off-centered sample disappears. Above 300 T, data for the centered sample is shown. In Fig. 4e, one can see the absence of the negative slope which is observed in Fig. 4f and 4g. This shows that the γ phase is absent at 185 K. On the other hand, we note that the transition corresponding to B_{C2} is observed even at 185 K, which is indicated by a star-shaped symbol in Fig. 4i. Thus, the upper boundary of the β phase is shown to be beyond 185 K.

We described above argument in the main text as follows.

We note that the sharpness of the transition at B_{C3} is lost in the $T_{ini} = 108$ K data shown in Fig. 3f. Furthermore, the negative slope itself disappears at the $T_{ini} = 185$ K data as shown in Fig. 3e, while the feature of the transition of B_{C2} remains. These features indicate that both β and γ phases become unstable at ~ 100 K, that the γ phase is completely absent at ~ 185 K, and also that the feature of the β phase does not completely disappear at ~ 185 K.

the revised part of the main text that appears in page 3

New Fig. 3a-3k, with the additional experimental result at 185 K.

Comments from reviewer #2 - 203

2) The measurement results are combined with a mean-field calculation (in comparison to previous model calculations, e.g. Sotnikov et al. *Sci. Rep.* 6 (2016) 30510 extended). In addition to the well-known phases SSO and EC (spin-state order or exciton condensation), new phases are postulated (ES becomes ES1 and ES2, possibly connected to β and γ) and shown by spin numbers in Fig. 4. The energy curves in Fig. 4, however, show only the states $|0.0\rangle$, $|0.2\rangle$ and $|2.2\rangle$ as states of the lowest energy. But, this field dependent scenario would be already known.

Our Reply

As shown below, Figs 4a-4c have been updated. The calculated mean field energy E_{MF} is shown with the orange-dashed line, which is lower than any of the known classical state $|0.0\rangle$, $|0.2\rangle$, and $|2.2\rangle$. Previously, the figure was so compressed that the difference is not apparent. We have now vertically enlarged the figure and changed the line style and thickness so that our present calculated result E_{MF} is apparent.

Before

Revised

Corrected part of Fig 4b is shown above

Comments from reviewer #2 - 204

3) *Why the transition from EC1 to EC2 should be of second order, as later discussed in the text, is also not clearly justified.*

Our Reply

We omit this argument from the text.

The finding of the second-order transition is the result of the calculation, not the experiment.

EC1's order is characterized by the nonzero order parameter of $\langle \tau^+ \rangle \neq 0$ and $\langle \rho^+ \rangle \neq 0$, a spatially uniform exciton condensation. In the EC2 phase, $\langle \tau^- \rangle$ and $\langle \rho^- \rangle$ are additionally nonzero, which causes translational symmetry breakings of exciton number per site. Thus, EC1 and EC2 are allowed to be connected via second-order phase transition.

Experimentally, we observed that continuous changes seem to occur between EC1 and EC2, that is, there is no sudden jump of ΔL . However, we admit that this is not conclusive because the B field sweep is very fast.

Comments from reviewer #2 - 205

4) The assignment of the calculated phases LS, EC1 and EC2 to α , β and γ and the real magnetic field to the "normalized field" seems arbitrary ("may correspond", "we speculate"). Some more concrete (quantitative) results of the simulation would make the connection between Figs. 3 and 4 more clear.

Our Reply

We revised Figs 4a-4c so that the horizontal axis has a connection to the real magnetic field. The horizontal axis of Figs 4a-f has been magnetic fields normalized to E_1 . The fact is not apparent in the previous version. Now, we have clearly indicated the label of the horizontal axis " h / E_1 " in the revised figure as shown below. Besides, we now use $(g_1, g_2) = (2, 4)$ instead of the previously used $(1, 2)$ because of the physical correspondence to the IS and HS states.

The bottom part of revised Figures 4b

In the mean-field calculation, E_1 is taken to be unity, which is the energy gap between vacuum (LS) and isolated exciton (IS) states at $B = 0$. When we use the value of E_1 to be 200 meV which is estimated for the CoO6 cluster in Ref. [M. W. Haverkort et al., Phys Rev Lett 97, 176405 (2006).], the reading of horizontal axis of $h / E_1 = 0.1$ corresponds to 20 meV. Then, with $g = 2$, $S_z = 1$, and $\mu_B = 5.79e-2(\text{meV/T})$, we obtain the value of $B = 172.7 \text{ T}$ for $h / E_1 = 0.1$ based on the relation of Zeeman energy. In Fig. 4b, the transition fields to EC1 and EC2 are 117, 263 T, respectively. In Fig. 4c, the transition fields to EC1 and EC2 are 73, 307 T, respectively. As an order estimation, the calculation in Figs. 4b and 4c show reasonable correspondences to the experimental result.

This argument is now described in the p6 of main text as follows.

the α phase. Here, we tentatively compare the transition fields of the experiment and the calculation. E_1 , the energy of isolated IS state, is reported to be ~ 0.2 eV with a CoO_6 cluster calculation in Ref. [41], which is assumed to be unity in the calculation. Then, we obtain $h/E_1 = 0.1 \rightarrow 172.7$ T as a scale of the horizontal axis of Figs. 4a-4c, with the Zeeman energy of $E = g\mu_B B S_z$. We obtain the transition fields of (161 T, 193 T, 432 T), (117 T, 263 T, 444 T), and (73 T, 307 T, 418 T, 485 T) for Figs. 4a-4c. As an order estimation, the obtained transition fields well account for the experimental result. The

The revised part of the main text which appears in the left column of page 6.

Fig4a-4c and the correspondence to the real magnetic field are shown

Comments from reviewer #2 - 206

5) *Some formal errors are: Fig. 3(d) does not exist. "Isotoropic" on page 4.*

Our Reply

Corrected.

Comments from reviewer #2 - 207

Overall, I have to come to the conclusion that the results presented are very valuable in the context of the long-lasting discussion due to the new suggestions and ideas and, especially the experimental results, should definitely be published. However, it must also be stated that the proposed conclusions bring only little clarification of the basic problem and do not resolve the existing controversies (even the possible LS-IS- HS hybridizations already discussed are not clarified any more than elsewhere).

*The manuscript is clearly written and linguistically correct, the illustrations instructive and of very good quality. However, the connection between experiment and theory does not yet seem to be fully harmonized. Because of this fact and because I cannot see enough significant new aspects for clarification on the basic problem of the spin-crossover in LaCoO₃, I do not want to recommend the publication in *nat.comm* at the present time.*

Our Reply

The first point “the connection between experiment and theory” is accomplished in the reply to #3-304. Here, we try to reply to the second remark. “enough significant new aspects for clarification on the basic problem of the spin-crossover in LaCoO₃”.

We have re-considered what is new and what we did to the long-standing problem of LaCoO₃. The summary is described as follows.

The very recent situation of the LaCoO₃ study:

- The HS-IS duality is claimed by Tomiyasu et al. in 2018 [K. Tomiyasu et al., arXiv:1808.05888.]. It is an important new idea that explains why LS-IS-HS hybridization occurs very simply from a microscopic perspective. Also, it explains why people sometimes observe HS and sometimes observe IS depending on measurements. Tomiyasu claimed the HS-IS duality to explain the result of inelastic neutron scattering that indicates the existence of ferromagnetic 7-site clusters. This idea is further verified with a study by Hariki using model calculation and resonant inelastic x-ray scattering [A. Hariki PRB (2020)]. Further experimental verification has been difficult because no ordered phase appears in LaCoO₃, except for those that appear under high magnetic fields. With the observation of only short-range order, we can not be sure whether the state mixing is due to the thermal effect or the quantum effect.
- In the meantime, spin-triplet exciton condensation is claimed for LaCoO₃ at high magnetic fields to account for the field-induced phases. However, for the sake of simplicity, the previous theoretical studies have concentrated on the mixing of the LS and the IS states, neglecting the HS state [A. Sotnikov and J. Kunes Sci. Rep (2016), PRB (2017), and T. Tatsuno et al., JPSJ (2016)]. As a result, those studies showed exciton condensation at high magnetic fields with only one kind of exciton condensation with increasing magnetic fields.

This study showed:

- The present experiment indicates multiple field-induced abnormal phases (β and γ phases), which are possible exciton condensations. They can not be explained with conventional field-induced EC theories with only LS-IS considered because they show only one kind of exciton condensation with increasing magnetic fields. It motivated us to include the HS state in the model calculation, which has never been done before.
- In the present theoretical calculation with the HS-IS duality term, we have succeeded in finding two kinds of exciton condensation with increasing magnetic fields, which is a new finding. The calculated result qualitatively reproduced the experimental observation of two possible exciton condensations under magnetic fields thanks to the inclusion of the HS-IS duality term.

Thus, this study has new indications:

- It is crucial to consider the HS-IS duality in LaCoO₃. It is supported for the first time based on the stabilization of the ordered phase, not in the disordered phase. We observed two candidates for exciton condensation (β and γ) which are not explained without the HS-IS duality. As a general message, our study suggested that the HS-IS duality is a strong candidate for the origin of the controversy of LaCoO₃, where people see both features of IS and HS in this material.

The above argument is now described in the main text as follows.

Here, we consider the present results in light of the spin-crossover physics in LaCoO₃, which has been a long-standing controversy for over half a century regarding the microscopic understanding of the temperature evolution of spin-states. To account for the peculiar spin-crossover in LaCoO₃ the involvement of various spin-states LS, IS, and HS, and also their interactions have been proposed [43]. With decreasing temperatures, LaCoO₃ transforms from a paramagnetic metal to a paramagnetic insulator at 500 K, and to a non-magnetic insulator at 100 K. As a possible inherent interaction, the attractive HS-LS interactions are proposed to account for the intermediate temperature range [44]. Later, involvement of IS state is proposed [45], and both supported and disputed by many studies [36, 46]. More recent studies indicate that the duality of IS-IS state and LS-HS state may be the origin of the intermediate temperature range, and also the source of the controversy as to whether IS or HS-LS is in play [38, 39]. The problem is that no long-range order has been reported in the temperature

evolution in spite of the proposed strong interactions [33, 47]. It is difficult to judge whether the state-mixing is due to the thermal effect or the quantum effect only with the observation of the short-range order in the temperature evolution. On the contrary, at ultrahigh magnetic fields, various ordered phases emerge. Thus, we can use high magnetic field phases for verification of the proposed interactions in LaCoO₃.

In the present study, the correspondence of the β and the γ phases to EC1 and EC2 phases indicates that the idea of HS-IS duality plays a critical role in the formation of the high magnetic field phases in LaCoO₃, because the co-appearance of EC1 and EC2 phases with increasing magnetic fields appear only when we include the $\mathcal{H}_{\text{dual}}$ term. It is also indicated by the appearance of ES phase with a small variation of V in the model calculation and its correspondence to the α phase that the delicate balance of the parameters of t , V , and J_{02} is in play in LaCoO₃. Thus, we argue that, in addition to the delocalization of IS state, the HS-IS duality and the competing parameters are of crucial importance when considering the controversial properties of LaCoO₃. We note our study is the first to support this aspect by observing the long-range ordered phase. Previous studies have focused on the thermally activated phases, where there is difficulty in judging whether the LS-IS-HS hybridization is a thermally activated one or is a quantum effect. More detailed natures of the exciton condensates in the β and the γ phases of LaCoO₃, will be clarified by future experiments and theories that focus on their magnetic nature, such as spin-transport or magnetic inelastic neutron scattering.

The revised part of the main text in page 6

Comments from reviewer #3 - 301

The manuscript experimentally investigated the magnetostriction of LaCoO₃ under magnetic fields as high as 600 T, with the aim of finding evidence for spin-triplet exciton condensation. The authors also proposed a theoretical model to further support their findings. However, the data presented do not convince me that they have achieved this goal. Therefore, I cannot recommend its publication in Nature Communications, at least not in its current version. My concerns are as follows:

Our Reply

We thank you for this criticism.

Comments from reviewer #3 - 302

1)Where is Fig. 3d? Without it, I cannot understand the results shown in the paragraph spanning pages 3 to 4 at all.

Our Reply

"Fig. 3d" was a typological error for "Fig. 3b". They are now corrected.

In particular, what does "gapless magnetostriction slopes" mean?

Our Reply

It is "a magnetostriction slope". "Gapless" is omitted.

How did the authors arrive at "These gapless behaviors evoke the Bose-Einstein condensation of spin-triplet excitons"?

Our Reply

In Bose systems, the expectation values of a creation operator b^+ and the compressibility should be nonzero when BEC occurs. The latter corresponds to the magnetic susceptibility in the present system with spin-triplet excitons.

In LaCoO₃, the continuous increase of exciton number is an indication of exciton BEC, while a plateau of exciton number indicates an exciton solid. The exciton number is directly correlated with the lattice volume. Thus, our measurement of ΔL indirectly senses the exciton number changes as a function of the external magnetic field. The continuous increase of ΔL indicates the Bose-Einstein condensation of spin-triplet excitons".

This is explained in more detail in the next Reply.

Comments from reviewer #3 - 303

2) Figure 2c shows the theoretical correspondence of the experimental work. The authors expected to verify the exciton condensate from observing such "Slope" and "Plateau" behaviors. Given the importance of this theoretical basis and the distinctly different backgrounds of the readers of Nature Communications, it would be better for the authors to elaborate on its relationship with the spin-triplet exciton condensation in more detail, rather than simply provide three references [14, 15, 26].

In my opinion, this hinders the readability and continuity of the manuscript. What makes me more confused is that there seems to be no work on this aspect in the references [14, 15, 26]. The word magnetostriction did not appear in any of the three publications. I wonder if the authors submitted the wrong version of the manuscript?

Our Reply

Thank you for this criticism.

In the present study, we have tried to measure the number of excitons as a function of external magnetic fields, instead of experimentally measuring the order parameter of exciton condensation, which is known as impractical. We have been inspired by the theoretical papers, where the evolution of the number of excitons as functions of external and internal parameters are reported. In Refs. [14, 15, 26], they report the change of the number of excitons as a function of external magnetic fields. As the reviewer pointed out, it is not a magnetostriction study. But, we regard that the number of excitons is firmly correlated with magnetostriction as follows.

In the current system of LaCoO_3 , the number of excitons is directly connected to the lattice volume. It is because the higher spin-states possess the larger ionic volume, thanks to the electrons occupying more extended orbitals of e_g . For these reasons, we resorted to the magnetostriction measurement up to 600 T using our state-of-the-art magnetostriction monitor, in order to follow the evolution of exciton number as a function of magnetic fields.

magnetostriction in the μ s-pulsed 1000 T environment. Magnetostriction is a crucial probe of the exciton states because the density of excitons is coupled to the lattice volume of LaCoO_3 . As indicated in Refs. [14, 15, 26], the plateaux and slopes in the magnetostriction correspond to solidifications and Bose-Einstein condensation of excitons in LaCoO_3 , respectively [See Fig. 2(c)]. Note that the correspondences are

Revision

pulsed 1000 T environment. Magnetostriction is a crucial probe of the exciton states because the density of excitons is coupled to the lattice volume of LaCoO_3 , where exciton and bi-exciton states have larger ionic volumes with the occupation of the more extended of e_g orbitals than t_{2g} orbitals as can be seen in the correspondence of spin-states and exciton states [See Figs. 1b and 1c.] As indicated in Refs. [20, 21, 24], the plateaux and slopes of exciton density correspond to solidifications and Bose-Einstein condensation of excitons, respectively. Thus, in LaCoO_3 , we can regard plateaux and slopes of magnetostriction as solidifications and Bose-Einstein condensation of excitons, respectively, which is schematically shown in Fig. 2c. Note that the correspondences are analogs to the magne-

The revised part of the main text that appears in page 2

Comments from reviewer #3 - 304

3) In the last part, the authors wrote down a complex Hamiltonian with eight parameters. Since the model does not include temperature explicitly, it is natural that it should be more suitable for the low temperature case like the α phase. But, as the authors pointed out, it fails to reproduce the α phase. So how do we understand the physics that this model gives us? With eight parameters, how to tell whether it is the result of tunable parameters or it really captures the essence of physics?

Our Reply

As reviewer #3 pointed out, the 8 parameters are enough for readers to get lost in seeing what is important.

We here try to decrease and rationalize the parameters. First, two parameters among them are g-factors, which are simply taken to be 2 and 4 for IS ($|1\rangle$) and HS ($|2\rangle$). Because we do not control them, we can omit them from the “control parameters”. The remaining 6 parameters are E_1 , E_2 , J_{11} , J_{02} , t , and V . They are the key parameters.

[M. M. Altarawneh et al, PRL (2012)]

$$\mathcal{H}_{\text{eff}} = J_1 \sum_{\langle ij \rangle} \sigma_i \sigma_j + J_2 \sum_{\langle\langle ij \rangle\rangle} \sigma_i \sigma_j - h \sum_i \sigma_i$$

- Classical Ising model
- Magnetic field
- 2 states of HS and LS

- NNN interaction is omitted for simplicity
- Quantum terms are added

Our Hamiltonian

$$\mathcal{H} = \mathcal{H}_{\text{loc}} + \mathcal{H}_{\text{int}} + \mathcal{H}_{\text{trans}} + \mathcal{H}_{\text{dual}}$$

Same classical terms
+ Quantum terms (NEW)

- Quantum terms added
- Magnetic field
- 3 states of LS, IS, HS ($\gg 0$, 1, 2 excitons)
- rooted in two-orbital Mott-Hubbard model

3 quantum states

~LS	~IS	~HS
Vacuum (S = 0)	Exciton (S = 1)	Bi-exciton (S = 2)

E_1 E_2

t $J_{11} \dots J_{02}$

Transfer

Interaction

Duality

V

We utilize 3 quantum states, which correspond to LS, IS, and HS states in LaCoO₃. The well-known Tanabe-Sugano diagram rationalizes E_1 (IS) $>$ E_2 (HS), which are taken to be 1.0 and 0.7, respectively. J_{11} and J_{02} are attractive interactions between IS-IS and LS-HS,

respectively. We use $J_{11} > J_{02}$ to stabilize the IS-IS pairs which are based on recent theoretical and neutron papers.

t and V are the quantum delocalization of IS to LS site, and HS resolving into IS-IS pair. Previously we show only one parameter. Now, we have shown in Figs. 4 the variation of the parameter V , which is the origin of the unique excitonic condensation in the present study, EC2.

As one sees in the revised Figs. 4 that, the EC1 and EC2 phase appears with $V = 0.17$, which presumably account for the β and γ phase. While, with $V = 0.14$, they almost disappear and an excitonic solid (ES) appears in a wide range of magnetic fields h . ES is one possible origin that is induced at low temperatures with changing parameters of V . On the other hand, $V = 0.2$ show a re-entrant to EC1 after EC2 is destroyed. Because we have not reached the magnetic saturation, we are not sure whether $V = 0.17$ or 0.2 is more appropriate to account for our study.

We note that, for the appearance of EC2, a sufficient amount of V is mandatory as seen in the figure below. Also, J_{02} needs to be sufficient. It is because EC2 is a hybridization between the HS-LS state and IS-IS state. For the appearance of EC1, a sufficient amount of t in addition to V is important because EC1 state is a hybridization of the LS, IS and HS states. In the present calculation, V is the most important parameter, which needs to be larger than twice the value of t . A systematic study of the parameter space is a topic of our future study.

FIG. 4. **a-c** Energy (E/E_1), exciton number (\bar{n}_i), and order parameters of exciton condensation ($\bar{\tau}$ and $\bar{\rho}$) of the many-body ground state in the proposed exciton model of LaCoO_3 calculated using the mean field approximation, shown as a function of external magnetic field (h/E_1). The calculated mean field energies (E_{MF}) are shown along with those of the isolated 2-site quantum states, $\langle E_{mn} \rangle = \langle m, n | (H_{000} + H_{100}) | m, n \rangle$ with $m, n = 0, 1, 2$. The duality parameter V responsible for $|2, 0\rangle_{ij} \rightleftharpoons |1, 1\rangle_{ij}$ is varied as (a) 0.14, (b) 0.17, (c) 0.2. See the main text for the details of the parameters. **d-g** Schematic illustrations of calculated many-body phases. (d) LS vacuum state, (e) Exciton condensation 1 (EC1), where Bose-Einstein condensation of excitons occur spatially uniformly, (f) Exciton condensation 2 (EC2), where Bose-Einstein condensation of excitons occur with a translational symmetry breaking, (g) HS: full saturation of exciton state with bi-exciton.

The revised figure 4 which appears in page 4

Revised Fig. 4a-4g are shown above

Above arguments are now described in the main text as follows

the $|2\rangle$ state and the $|1\rangle$ state [38, 39]. \mathcal{H}_{loc} and \mathcal{H}_{int} are the classical terms which are introduced in the classical exciton model in Ref. [28]. $\mathcal{H}_{\text{trans}}$ and $\mathcal{H}_{\text{dual}}$ are the quantum terms which are responsible for hybridization and exciton condensation. Especially, $\mathcal{H}_{\text{dual}}$ is inspired by the recent idea of the HS-IS duality introduced in Refs. [38, 39], which has not been considered in the previous calculations on the exciton condensations of LaCoO_3 under ultrahigh magnetic fields [20, 21]. Note that the next-nearest-neighbor interactions are considered in Ref. [28], which are neglected in the present model for simplicity. In the calculation, the duality parameter V is varied as 0.14, 0.17, 0.2. The other parameters are chosen to be $(E_1, E_2, J_{11}, J_{02}, t) = (1.0, 0.7, 0.2, 0.05, 0.07)$. The condition $E_1 > E_2$ is rationalized by Tanabe-Sugano diagram for the octahedrally coordinated d^6 electrons [40]. $J_{11}, J_{02} > 0$ stabilize the states $|1, 1\rangle$ and $|0, 2\rangle$ with the attractive interactions [28]. $t > 0$ switches on the itineracy of $|1\rangle$ state [14]. In the calculation, the larger value of V than t has been significant to obtain the present results.

The revised part.

This paragraph appears in the left column of page 6

As reviewer #3 pointed out we did not succeed in reproducing α phase in the previous model calculation, despite the fact α phase is the low-temperature phase and that the model is within the zero temperature limit. As you see in the revised Fig. 4a-4c, we have shown the variation of the parameter V which describes the HS-IS duality. It results in the appearance of the non-trivial phases of ES, EC1, and EC2, which are candidates for α , β , and γ phases. By the variation of the HS-IS duality parameter V , we have succeeded in reproducing ES phase as an origin of the α phase in the model.

ES appeared in the different parameter of V ($= 0.14$) from those ($V = 0.17$ and 0.2) for the appearance of EC1, and EC2. It is due to the situation that the α phase and β , and γ phases have distinct lattice parameters, resulting in distinct interaction parameters. We suspect that, in the α phase, the electron hopping is reduced in t_{2g} orbitals, resulting in the enhancement of the repulsive interactions for excitons so that more localized phases like super-lattices are formed in the α phase. This should be the origin of the plateau behavior of the magnetostriction in the α phase.

The change of interaction parameters in α and β , γ phases should have happened with the strong lattice expansion in $\alpha \rightarrow \beta$ transition. Such lattice shrinkage is actually reported in the paper [A. Ikeda et al., Phys. Rev. Lett. (2020)] with increasing temperature with constant high magnetic fields ~ 100 T, which is shown below.

Further analysis of the changing interaction parameters with regard to the lattice changes is an interesting topic of research. One needs to take into account the bond angle, length, and symmetry change. Although it is beyond the scope of the present paper, we are now collecting x-ray diffraction data in the α phase, based on the state-of-the-art technique utilizing a portable ultrahigh magnetic field generator [A. Ikeda et al., Appl. Phys. Lett. (2022)].

We note this point in the main text as follows.

the lattice volume in EC2. We argue that the α phase emerges with modified interaction parameters as compared to those for the β and γ phases, as ES appears with the variation of the HS-IS duality parameter V suppressing the appearance of EC1 and EC2. The modification of the parameter should occur due to the strong lattice contraction, which enhances the localization nature of excitons, resulting in exciton superlattice formation in the α phase. Such lattice contraction in the course of the β - α phase transition above 100 T is actually observed in Ref. [16]. Further microscopic understanding of the lattice change affecting the interaction parameters will be an interesting work in the future which will be accomplished using the state-of-the-art technique utilizing an x-ray free electron laser and a portable generator of ultrahigh magnetic fields [42].

The revised paragraph.

This paragraph appears in the left column of page 6

Comments from reviewer #3 - 305

4) *There are several typos.*

Our Reply

Thank you.

The manuscript is thoroughly reviewed for typos and expressions. The linguistic revision is indicated by the blue-colored text in the main manuscript.

REVIEWER COMMENTS

Reviewer #1 (Remarks to the Author):

The authors have carefully revised the article and given exhaustive comments on referees' remarks. There still remains room for criticism on some points (sample vibrations and correspondence between the experimental data and theoretical interpretation).

Nevertheless, I recommend accepting the revised article for publication for the following reasons. The authors applied the novel promising experimental technique (magnetostriction measurements in flux compression generator) with the best accuracy currently possible. This has allowed to explore the phase diagram of LaCoO₃ in ultrahigh magnetic field and interpreted it in terms of Bose condensation. The work can become the basis for a new line of research and is of interest to a wide scientific audience.

Reviewer #2 (Remarks to the Author):

Referee report to

Authors:

A. Ikeda, Y.H. Matsuda, K. Sato, Y. Ishii, H. Sawabe, D. Nakamura, S. Takeyama, and J. Nasu

Title:

“Spin triplet condensations in LaCoO₃ at ultrahigh magnetic fields up to 600 T”

The authors have comprehensively and very thoroughly dealt with the issues raised in the process of refereeing, leaving themselves plenty of time for reflection and thorough discussion. For this they are comprehensively thanked in these fast-moving times!

The manuscript has been fleshed out, supplemented and expanded, especially in the explanation of the theoretical model. But, in my opinion, this extension (now 8 instead of 6 pages) does not make the statements any clearer:

-For example, see fig. 4, which now contains parts a-c in graphic form. The difference results from different values for the duality parameter V . But how does this selection come about? On the other hand, the delocalisation parameter t was not varied. Where does the value 0.07 come from ?

-Further, following the questions of the referees, absolute values of the magnetic field for the transitions are given, but they do not fit the experimental findings (even if it is clear that it is only an "estimation").

-Very helpful is the insertion that excitons and bi-excitons lead to a larger volume (full agreement !). Only here it was not dV/V but dL/L that was measured, both do not have to change in the same way. So the fit of the experimental anomalies with the calculated transitions based on this remains very fragile. The expected calculation or at least estimation of magnetostriction from the model does not become clear !

In summary: As already stated, the experimental work is great. Also the duality idea HS-IS founded by the extended theoretical model combined with a dynamical exciton condensation as a basic idea and thus a much more complex system instead of the LS-HS OR IS-IS - scenarios considered at the beginning is interesting. But: Both do not yet fit together sufficiently!

Far be it from me to set aside these two fundamental works.

As a compromise, I strongly recommend the authors to present the experimental material by all means, but only to name the basic ideas of the theory, including the Hamiltonian, and to shift the details of the calculation to the supplemental (instead of the statements on the well known generation of megagauss fields and the measurement of magnetostriction).

With this, I would recommend the present manuscript for acceptance with the proposed major editorial changes.

Reviewer #3 (Remarks to the Author):

In the reply, the authors further explain the experiment and refine the theoretical model. However, these do not help to fully address the central matter of my previous comment: whether the authors indeed observed spin-triplet exciton condensation and how the condensation was confirmed from the magnetostriction technique used here. In particular, it is necessary to provide clear evidence to rule out other hypotheses, such as spin-state order. But instead of answering my question directly,

the authors simply assume that "the number of excitons is directly related to the lattice volume", which completely ignores the essence of the problem: the relationship between ΔL and exciton condensation. I understand that they tried to use a theoretical model to help confirm this. But a model with 6 parameters can reproduce many things, not only exciton condensation, and I believe that adjusting the parameters can lead to other explanations as well. All the evidence provided does not convince me that they did observe spin-triplet exciton condensation, while just pointing to this possibility. Judging from the insufficient scientific advance, I cannot recommend its publication in Nature Communications.

Point-by-point response

Comments from reviewer #1 - 101

The authors have carefully revised the article and given exhaustive comments on referees' remarks. There still remains room for criticism on some points (sample vibrations and correspondence between the experimental data and theoretical interpretation). Nevertheless, I recommend accepting the revised article for publication for the following reasons. The authors applied the novel promising experimental technique (magnetostriction measurements in flux compression generator) with the best accuracy currently possible. This has allowed to explore the phase diagram of LaCoO₃ in ultrahigh magnetic field and interpreted it in terms of Bose condensation. The work can become the basis for a new line of research and is of interest to a wide scientific audience.

Our reply:

We really appreciate the supporting comment. We commented on the sample vibrations in the supplementary material. The theoretical parts are shortened and most of them are moved to the supplementary material.

Comments from reviewer #2 - 201

The authors have comprehensively and very thoroughly dealt with the issues raised in the process of refereeing, leaving themselves plenty of time for reflection and thorough discussion. For this they are comprehensively thanked in these fast-moving times! The manuscript has been fleshed out, supplemented and expanded, especially in the explanation of the theoretical model. But, in my opinion, this extension (now 8 instead of 6 pages) does not make the statements any clearer:

-For example, see fig. 4, which now contains parts a-c in graphic form. The difference results from different values for the duality parameter V . But how does this selection come about? On the other hand, the delocalisation parameter t was not varied. Where does the value 0.07 come from ?

Our reply:

Thank you. We have moved Fig.4 to the supplementary material. Besides, we have shown the calculated results with the variation of the parameter ' t ' in the supplementary material. This figure indicates that EC1 and EC2 appear in the wide parameter range for ' t '.

The new figures were added to supplementary material to discuss each phase's dependence on t , V , and h .

Comments from reviewer #2 - 202

-Further, following the questions of the referees, absolute values of the magnetic field for the transitions are given, but they do not fit the experimental findings (even if it is clear that it is only an "estimation").

Our reply:

We agree that our model show only quantitative agreements with the experiment. We still believe such discussion is important to show physics qualitatively. So we have moved these parts to the supplementary material.

Comments from reviewer #2 - 203

-Very helpful is the insertion that excitons and bi-excitons lead to a larger volume (full agreement !). Only here it was not dV/V but dL/L that was measured, both do not have to change in the same way. So the fit of the experimental anomalies with the calculated transitions based on this remains very fragile. The expected calculation or at least estimation of magnetostriction from the model does not become clear!

Our reply:

Thank you for the positive comment on the representation of the relation between the exciton states and the lattice volume. We comment on the difference between dV/V and dL/L . In the case with no spin-orbit interactions, dL/L is proportional to dV/V . Actually, such a situation is common in 3d transition metal oxides, where the orbital angular momentum (L) is usually quenched. In the case of LaCoO_3 , the orbital angular momentum is not completely quenched but small enough to be neglected in the present experiment, which is indicated by the almost isotropic g-factors ($g_{\parallel} = 3.35$, $g_{\perp} = 3.55$) in LaCoO_3 investigated by an electron spin resonance study [S. Noguchi, et al, Phys. Rev. B 66, 094404 (2002)]. This point is discussed in the supplementary material.

D. Consideration on spin-orbit interactions

We comment on the difference between dV/V and dL/L . Spin-state originates purely in the electron spin (S) when no spin-orbit interaction ($S \cdot L$) is in play. In that case, dL/L can represent dV/V . Actually, such a situation is common in transition metal oxides, where the orbital angular momentum (L) is usually quenched. In the case of LaCoO_3 , the orbital angular momentum is not completely quenched. Thus, an influence of spin-orbit interaction may result in $dV/V \neq dL/L$ in magnetostriction even for the polycrystalline sample used in the present study. In reality we may ignore this because it is reported small enough to be neglected in the present experiment by an electron resonance study showing the almost isotropic g-factors ($g_{\parallel} = 3.35$, $g_{\perp} = 3.55$) in LaCoO_3 [S. Noguchi, et al, Phys. Rev. B 66, 094404 (2002)].

We comment on another experimental fact indicating that we can ignore the spin-orbit couplings. Previously, we argued that the magnetization increases with increasing magnetic fields in β phase although magnetostriction is a constant, which may induce the spin-driven lattice change through spin-orbit coupling. We confirmed that this is not the case with measurements of magnetostriction and magnetization at room temperature, where only magnetization increases up to 50 T without any increase in magnetostriction [3]. It is reasonable to assume that the spin-orbit coupling, a local interaction, is temperature independent and is also absent at 78 and 108 K. This justifies our assumption that the longitudinal magnetostriction is proportional to lattice volume in the present study. The influence of orbital order is also unlikely either because we presently used a polycrystalline sample.

Comment on spin-orbit interaction in the supplementary material. Blue colored part is newly added.

Comments from reviewer #2 - 204

In summary: As already stated, the experimental work is great. Also the duality idea HS-IS founded by the extended theoretical model combined with a dynamical exciton condensation as a basic idea and thus a much more complex system instead of the LS-HS OR IS-IS - scenarios considered at the beginning is interesting. But: Both do not yet fit together sufficiently!

Far be it from me to set aside these two fundamental works.

As a compromise, I strongly recommend the authors to present the experimental material by all means, but only to name the basic ideas of the theory, including the Hamiltonian, and to shift the details of the calculation to the supplemental (instead of the statements on the well known generation of megagauss fields and the measurement of magnetostriction).

With this, I would recommend the present manuscript for acceptance with the proposed major editorial changes.

Our reply:

Thank you for thinking highly of our results and also for the suggestion of the edition. We followed your guidance to edit the main text. Specifically, the details of the calculations are moved to the supplementary material and only the outline of the model calculation is described in the main text. Now the main text is concise and 6 pages, which was 8 pages in the previous version.

Overall view of the edition made to the latter half of the manuscript. The blue-colored part in the present version summarizes the theoretical part.

Comments from reviewer #3 - 301

In the reply, the authors further explain the experiment and refine the theoretical model. However, these do not help to fully address the central matter of my previous comment: whether the authors indeed observed spin-triplet exciton condensation and how the condensation was confirmed from the magnetostriction technique used here. In particular, it is necessary to provide clear evidence to rule out other hypotheses, such as spin-state order. But instead of answering my question directly, the authors simply assume that "the number of excitons is directly related to the lattice volume", which completely ignores the essence of the problem: the relationship between ΔL and exciton condensation. I understand that they tried to use a theoretical model to help confirm this. But a model with 6 parameters can reproduce many things, not only exciton condensation, and I believe that adjusting the parameters can lead to other explanations as well. All the evidence provided does not convince me that they did observe spin-triplet exciton condensation, while just pointing to this possibility. Judging from the insufficient scientific advance, I cannot recommend its publication in Nature Communications.

Our reply:

We appreciate this criticism. We realize that the heart of the concern of reviewer #3 is "the relationship between ΔL and the order parameter of exciton condensations". In other words, "Is the order parameter of exciton condensation observed in the present study through magnetostriction?" The answer is NO. To be precise, the order parameter of exciton condensations is the global alignment of the phase factor of the wavefunction of every exciton. In theoretical calculations, this can be evaluated as the expectation value of the creation or annihilation operator of boson, $\langle b^+ \rangle$, because the number of condensed boson is related to the phase of wavefunction with the uncertainty principle. However, in experiments, the coherence of exciton condensation is invisible to any kind of measurements. Usually, experiments look at the resultant macroscopic or microscopic phenomena and resort to the process of elimination. Our study is also an example of such a study.

As the reviewer #2 stated, it is sensible that the exciton occupation state is related to the lattice volume because the exciton and bi-exciton states have larger volumes than the LS vacuum state without electrons occupying eg orbitals. Experimentally, we use this information as a clue to the exciton condensation. It is not the direct evidence. Now we clearly state this fact in the manuscript.

Since ΔL is not directly related to exciton condensation, we focus on the magnetic field dependence of ΔL , which is related to the exciton density, in our macroscopic experiments. As stated in the manuscript, the exciton density shows a plateau in the classical states with a periodic alignment of excitons, such as exciton crystal (spin-state crystal) and fully-occupied bi-exciton state (field-induced HS state). Namely, these states show magnetostriction plateaux at low temperatures. On the other hand, in the case of exciton condensation, the exciton density changes by changing the magnetic field even without thermal fluctuations. This difference provides a clue to identifying exciton condensation. In the β and γ phases, ΔL depends on the magnetic field with showing slopes, unlike the case in the α phase. Furthermore, the bending behavior becomes sharper and more apparent with decreasing temperature, implying that the slope structures survive even without thermal fluctuations and

originate from quantum effects. These results rule out the possibility of spin-state crystals and suggest that the magnetostriction slopes observed here are ascribed to the quantum mixing of distinct exciton states. Therefore, exciton condensation is the most plausible to understand the experiment for the β and γ phases. Hence, we state that the experimental observation of the nonthermal slopes (continuous changes) of the magnetostriction is the findings in the present study which we believe are the signatures of the exciton condensations. To our knowledge, there is no other simple explanation to account for the slope of magnetostriction.

indicating a new phase transition at 380 T denoted as B_{C3} from the β phase to a novel state in the data at $T_{ini} = 78$ K. The new state is termed γ phase, which is characterized by the positive slope beyond 380 T [See slope 2 in Fig. 3g]. The sharp slopes of slope 1 and slope 2 at $T_{ini} = 78$ K become smeared in the data obtained at $T_{ini} = 108$ K as shown in Fig. 3f. Thus,

the sharp slopes of slope 1 and slope 2 obtained at $T_{ini} = 78$ K are not thermal origins. But rather, we argue that they originate in quantum fluctuations of excitons such as exciton condensations as schematically shown in Fig. 2c. Previously in Ref. [16], the β phase was falsely considered a plateau due to the measurement regions limited below 200 T. Furthermore,

The revised part on page 3

On the other hand, the β and the γ phases show magnetostriction slopes denoted as slope 1 and slope 2 in Figs. 3f and 3g. Based on the observation that they are nonthermal in origin, they indicate that exciton density varies continuously as a function of magnetic fields. Considering the firm cou-

The revised part on page 4

Please note that our experiment is not an exception to the well-known notion that no experiment can directly confirm the order parameter of exciton condensation, which is the alignment of the phase factor of the wavefunctions of every exciton. Any resultant macroscopic or microscopic phenomena can not be a direct probe. For instance, the resultant change of the charge gap is observable. However, it can also occur as a result of other phenomena like lattice distortions which are not directly related to the quantum coherence of the exciton condensation.

We agree with the reviewer's opinion that our experiment is not compelling evidence of the spin-triple exciton condensation. So we decide to soften our tone for the conclusion. We state that we observed the signature of exciton condensations, which is the nonthermal slopes of the magnetostriction.

We agree with the reviewer on the suggestion that the previous manuscript sounds too *conclusive*, while our experimental result and calculation are only *suggestive*. In the revised manuscript, we have weakened our tone as to how the discussion is conclusive. Representatively, we have added the word "Signature of" on top of the title of our manuscript. We have colored the revised text in magenta color regarding these changes.

Last but not least, we emphasize that our work established a new technique to observe electronic states in extreme environments under strong magnetic fields. We are confident that the present study significantly advances the science of transition metal oxides and high magnetic fields. Despite the softened expressions, our work remains quite a significant scientific advance in the scientific fields of transition metal oxides and high magnetic fields.

Signature of spin-triplet exciton condensations in LaCoO₃ at ultrahigh magnetic fields up to 600 T

Akihiko Ikeda,^{1,2,*} Yasuhiro H. Matsuda,¹ Keisuke Sato,³ Yuto Ishii,¹
Hironobu Sawabe,¹ Daisuke Nakamura,¹ Shojiro Takeyama,¹ and Joji Nasu^{4,5}

¹*Institute for Solid State Physics, University of Tokyo, Kashiwa, Chiba 277-8581, Japan*

²*Department of Engineering Science, University of Electro-Communications, Chofu, Tokyo 182-8585, Japan*

³*National Institute of Technology, Ibaraki College, Hitachinaka, Ibaraki 312-0011, Japan*

⁴*Department of Physics, Tohoku University, Sendai, Miyagi 980-8578, Japan*

⁵*PRESTO, Japan Science and Technology Agency, Honcho Kawaguchi, Saitama 332-0012, Japan*

(Dated: December 26, 2022)

Bose-Einstein condensation of electron-hole pairs, exciton condensation, has been effortfully investigated since predicted 60 years ago. Irrefutable evidence has still been lacking due to experimental difficulties in verifying the condensation of the charge neutral and non-magnetic spin-singlet excitons. Whilst, condensation of spin-triplet excitons is a promising frontier because spin supercurrent and spin-Seebeck effects will be observable. A canonical cobaltite LaCoO₃ under very high magnetic fields is a propitious candidate, yet to be verified. Here, we unveil the exotic phase diagram of LaCoO₃ up to 600 T generated using the electromagnetic flux compression method and the state-of-the-art magnetostriction gauge. **We found the continuous magnetostriction curves and a bending structure, which suggest the emergence of two distinct spin-triplet exciton condensates. By constructing a phenomenological model, we showed that quantum fluctuations of excitons are crucial for the field-induced successive transitions.** The spin-triplet exciton condensation in a cobaltite, which is three-dimensional and thermally equilibrated, opens up a novel venue for spintronics technologies with spin-supercurrent such as a spin Josephson junction.

Revision to the title and abstract.

tion of e_g orbitals that is spatially more extended than t_{2g} orbitals as can be seen in the correspondence of spin-states and exciton states [See Figs. 1b and 1c.] **Solidifications and Bose-Einstein condensations of excitons result in plateaux and slope of exciton density [20, 21, 24]. Thus, we expect that they also result in plateaux and slope in magnetostriction curves at very low temperatures, which is schematically shown in Fig. 2c.** Note that the correspondences are analogs to the magnetization plateaux and slopes in magnon solids and superfluids, respectively [25, 26].

Revision to a paragraph on page 2.

Many thanks to the criticism from reviewer #3. Our manuscript becomes more sound from a scientific point of view. We really appreciate the criticism.

End of letter